# New Species and Records Expand the Checklist of Cellular Slime Molds (Dictyostelids) in Jilin Province, China

**DOI:** 10.3390/jof10120834

**Published:** 2024-12-02

**Authors:** Zhaojuan Zhang, Liang He, Yuqing Sun, Zhuang Li, Yingkun Yang, Chao Zhai, Steven L. Stephenson, Xiangrui Xie, Yu Li, Pu Liu

**Affiliations:** 1Engineering Research Center of Edible and Medicinal Fungi, Ministry of Education, Jilin Agricultural University, Changchun 130118, China; 17843098488@163.com (Z.Z.); 15844982685@163.com (L.H.); y3050644744@126.com (Y.S.); yangyingkun2020@163.com (Y.Y.); zhai13029119483@163.com (C.Z.); 18381823422@163.com (X.X.); fungi966@126.com (Y.L.); 2Shandong Provincial Key Laboratory for Biology of Vegetable Diseases and Insect Pests, College of Plant Protection, Shandong Agricultural University, Tai’an 271018, China; liz552@126.com; 3Science and Technology Research Center of Edible Fungi, Qingyuan 323800, China; 4Department of Biological Sciences, University of Arkansas, Fayetteville, AR 72701, USA; slsteph@uark.edu

**Keywords:** dictyostelids, biodiversity, new taxa, resource utilization

## Abstract

Dictyostelids represent a crucial element in the protist community, and their abundant presence in Jilin Province underscores their indispensable role in biodiversity conservation. In the present study, a resource survey of dictyostelids used random sampling to collect 28 soil samples from five localities in Changbai Korean Autonomous County, Jilin Province. In addition, a compilation of dictyostelid species reported from Jilin Province was developed, based on a thorough review of the literature. The survey yielded fifteen isolates of dictyostelids, comprising six species from four genera. Notably, two new species (*Dictyostelium longigracilis* sp. nov. and *Dictyostelium macrosoriobrevipes* sp. nov.) were described using morphological characteristics and SSU gene-based phylogenetic analyses. One other species (*Polysphondylium patagonicum*) was recorded as new for China, while another (*Cavenderia aureostipes*) was recorded as a new record for Jilin Province. The dictyostelid assemblage in Jilin Province is dominated by the genus *Dictyostelium* (51.4%), with a total of 35 species, which represent 59.3% of the current total known for all of China. These findings provide a scientific basis for the protection of species diversity and resource utilization of dictyostelids in Jilin Province.

## 1. Introduction

Species richness, specifically referring to the variety of plant, animal, and microbial species (hereafter referred to as ‘biodiversity’), is crucial to human existence as it underpins ecological balance, sustains natural processes, provides food, medicine, and numerous resources, and serves as a foundation for scientific research and esthetic appreciation [1,2,3]. For example, soil harbors a broad diversity of microorganisms that are well recognized drivers of key processes such as nutrient cycling and plant growth [4,5]. Just like aboveground animal and plant communities, the diversity and composition of soil-dwelling microbial communities can vary markedly across the globe [6]. As is known, protists actively participate in belowground ecosystems, both in absolute density (a single gram of soil contains tens of thousands or more individuals) and in the level of energy transfer they provide [7,8,9].

Dictyostelid cellular slime molds (dictyostelids or heterotrophic amoebae [10]) comprise a group of sorocarp-forming eukaryotic protists that inhabit the surface layers of soils worldwide [11,12,13]. However, they have received far less attention than other components of the soil microbiome. Due to their small size, their study is currently limited to laboratory culture [14]. As the second largest group of slime molds [15], dictyostelids encompass almost 200 described species, second only to the true slime molds (myxomycetes [16]). As such, these organisms are an important component of biodiversity. In addition, as bacterial predators, they potentially play important roles in soil ecology and promote soil and plant health by performing top–down control on ecosystem processes (e.g., decomposition) in which bacterial populations are involved. Since dictyostelids are essentially indistinguishable, their identification and taxonomy have traditionally been based on the morphology of their sorocarps and related characters, but this has recently been augmented by molecular phylogeny. They are particularly well known and widely studied for their lifestyle, which alternates between uni- and multicellular (sorocarpic) stages and represents one of nature’s tractable system to resolve the evolution of cell-type specialization and inventions of multicellularity [17,18].

In recent years, research on the diversity of dictyostelids and their environmental impact has attracted the attention of scholars in China and abroad. China covers a vast territory characterized by a large number of terrestrial slime molds (*Catalogue of Life China: 2024 Annual Checklist*, http://www.sp2000.org.cn, accessed on 12 July 2024; [19]). During the period from 1983 to 2024, 59 taxa of dictyostelids (58 species and 1 variety) from 2 orders and 4 families were reported from samples collected from 25 provincial administrative regions, covering 4 climate zones (tropical, subtropical, temperate, and alpine regions) [20,21,22]. Among these, Jilin is the province with the richest dictyostelid diversity due to its diverse topography, temperate continental monsoon climate, and abundant ecosystems. Extensive research has been undertaken to explore the diversity of dictyostelids within Jilin Province [21,23,24]. 

The classification system of dictyostelids has undergone considerable change in the past decade, and numerous families have been classified and revised [25,26,27]. The objective of this study was to systematically summarize and organize existing reports on species of dictyostelids from Jilin Province according to the latest classification system and subsequently conduct appropriate supplementary investigations based on this baseline of data.

## 2. Materials and Methods

### 2.1. Study Area

Changbai Korean Autonomous County is located in the southeast of Baishan City and southeast of Jilin Province, China (40°37′ N–41°05′ N, 127°17′ E–128°29′ E) (Figure 1a). Changbai Korean Autonomous County has a sub-frigid continental monsoon climate, featuring uneven temperatures in the spring, warm and rainy conditions in the summer, clear and crisp days in the autumn, and cold and long days in the winter. The rainfall is abundant and concentrated. The average annual precipitation is 605.8 mm, of which 374.2 mm falls from June to August, accounting for 57% of the annual precipitation. The average annual frost-free period is 123 d, the annual sunshine duration is 2458.7 h, the average annual wind speed is 2.1 m per second, and the most frequent wind direction is northeasterly. The annual number of days with thunderstorms is 37.4 (available online: http://changbai.gov.cn/cbgk/zrgk/201801/t20180108_257149.html; http://www.dajilin.com/mobile/node_6043.html accessed on 12 July 2024).

### 2.2. Sampling, Isolation, and Cultivation

Briefly, 28 soil samples were collected from five localities (L1–L5) in Changbai Korean Autonomous County, representing four vegetation types (coniferous forest, broadleaf forest, mixed forest, and farmland) in July 2021 (Table 1). L1 includes coniferous forest; L2 includes farmland and broadleaf forest; L3 includes mixed forest, broadleaf forest, and farmland; L4 and L5 include broadleaf forest (Figure 1a). Each sample consisted of 30 to 40 g from surface soil (0–10 cm in depth) based on a random sampling method and was placed in a sterile Whirl-Pak plastic bag. Within a few hours of collection, all samples were transported to a laboratory. The samples were numbered and recorded in the laboratory soil sample database and then stored at 4 °C.

The isolation protocols used in this research followed those described by Cavender and Raper [29] (Figure 1b). Each sample was weighed, and enough ddH_2_O was added for an initial dilution (1:10). This dilution was shaken to disperse the soil particles and suspend the amoebae, microcysts, and spores of dictyostelids. Afterward, an 0.5 mL aliquot of this dilution and 0.4 mL *Escherichia coli* (food source) were dispensed into three duplicate culture plates prepared with hay infusion agar. The plates were incubated at 23 °C with a 12 h light-and-darkness cycle. 

For a period of two weeks following the initial emergence of aggregations, each plate was carefully inspected at least once daily. Each isolate recovered from one of the plates was purified and cultivated for taxonomic studies and preservation on non-nutrient water agar plates with *E. coli* pre-grown for 12 to 24 h. Spores from these plates were frozen in a HL5 [30] medium and stored at −80 °C in the herbarium of the Mycological Institute of Jilin Agricultural University (HMJAU), Changchun, China.

### 2.3. Observation of Morphological Features

Morphological characteristics in the life cycle were described as proposed by Hagiwara [31] and Raper [12]. Under a stereoscopic microscope, isolates were observed and features of their morphology documented. This features included growth habit, branching, color, sorocarp characteristics, and sizes/shapes of aggregates and pseudoplasmodia. All these features were observed under a fluorescent stereo microscope (Leica M165FC; Leica, Wetzlar, Germany). In each instance, a viable and morphologically normal-appearing dictyostelid isolate, referred to as a “healthy” isolate, was selected under a dissecting microscope, mounted on a slide with sterile water, covered, and then observed under an optical microscope. The microscopic features observed were spore size/shape, polar granules, sorophore, cell rows, and sorophore tips/bases. These were observed using a Zeiss light microscope (Axio Imager A2; Zeiss, Oberkochen, Germany), with 10 ocular and 10, 40, and 100 (oil) objectives. Photographs were taken with a Zeiss AxioCam 506 color microscope camera.

We compared the recorded information and photographs of dictyostelids with the morphological characteristics documented in the original literature, which was referenced in Index Fungorum [32] (http://www.indexfungorum.org/names/Names.asp, accessed on 7 May 2022), to accurately identify the species isolated in the present study.

### 2.4. DNA Isolation, PCR Amplification, Sequencing, and Phylogenetic Analysis

The molecular methods employed in this study followed those described by Liu et al. [33]. The spores of each isolate were collected with a sterile tip and mixed with the lysis buffer of the MiniBEST universal genomic DNA extraction kit ver.5.0 (TaKaRa Bio Inc., Kusatsu, Japan) according to the manufacturer’s protocol. The genomic DNA solution was used directly for PCR amplification using the primers [34] 18S-FA (5′-AACCTGGTTGATCCTGCCAG-3′) and 18S-RB (5′-TGATCCTTCTGCAGGTTCAC-3′). The 25 μL reaction volume consisted of 12.5 μL Premix Taq™ (Ex Taq™ Version 2.0), 1 μL of each forward and reverse primer, 1 μL template genomic DNA (about 10 ng/μL), and 8.5 μL ddH_2_O. PCR products were visually inspected on agarose electrophoresis gels and compared with band intensities of 1600–1800 bp DNA ladders.

PCR products were sent to Sangon Bio-tech Co., Ltd. (Shanghai, China), for sequencing. Following thorough verification, the fifteen newly generated sequences were confirmed to be accurate and were subsequently submitted to GenBank. All sequences of related species were downloaded from GenBank for phylogenetic analysis to determine their phylogenetic relationships with other taxa in the group (Table 2).

All SSU sequences were aligned and compared separately using the ClustalW Multiple alignment [35], then manually adjusted in BioEdit version 7.0.9.0 [36]. Maximum-likelihood analyses (ML analyses) were performed using IQTREE v.1.6.12 [37], with 10,000 replicates of ultrafast-likelihood bootstrapping to obtain node support values with the “-bb 10,000” option and further optimized using a hill-climbing nearest-neighbor interchange (NNI) with the “-bnni” option [38]. We also used the SH-aLRT test to obtain the confidence limit of the topology with the “-alrt 1000” option [39]. The “-nt AUTO” option was used to automatically determine the best number of cores given the current data. We used ModelFinder as implemented within IQ-TREE to determine the best substitution model based on Bayesian information criteria (BIC) [40]. We used a myxomycete sequence (*Physarum polycephalum*, accession no. X13160.1/NC_002508.1) as the outgroup with the “-o X13160.1”/“-o NC_002508.1” option. The phylogenetic tree was beautified using iTOL [41] for display, manipulation, and annotation purposes.

## 3. Results

Fifteen isolates representing six species of dictyostelids were recovered from soil samples collected at four localities (L2–L5) in Changbai Korean Autonomous County, out of a total of twenty-eight samples gathered from five localities (Table 1, Figure 1). Four isolates (*Dictyostelium longigracilis* and *Dictyostelium macrosoriobrevipes*) (Figure 2 and Figure 3) from localities L2 and L5 were new to science. The species *Dictyostelium robusticaule* (Figure 4) from L2 had been reported from Jilin Province previously. Two isolates (*Polysphondylium patagonicum*) (Figure 5) from locality (L5) were new records from China. The species *Heterostelium candidum* (Figure 6) from L4 had been reported from Jilin Province previously. Seven isolates (*Cavenderia aureostipes*) (Figure 7) from localities (L3–L5) were new records for Jilin Province. One locality (L1) did not yield any isolates. Furthermore, phylogenetic studies of the ribosomal small subunit (SSU) further support the taxonomic placement of all six species recorded from the Changbai Korean Autonomous County (Figure 8, Figure 9, Figure 10 and Figure 11). All isolates and soil samples of the species considered herein were deposited in the mycological herbarium of Jilin Agricultural University.

### 3.1. Taxonomy and Molecular Phylogeny

***Dictyostelium longigracilis***, found by Z.J. Zhang, P. Liu, and Y. Li sp. nov., is shown in Figure 2.

MycoBank: FN 572069.

When cultured at 23 °C on non-nutrient agar with *Escherichia coli*, the **sorocarps** are solitary, erect, or inclined, sometimes prostrate on the medium, unbranched, strongly phototropic, with a relatively fast growth rate, reaching 10 mm in 48 h, fragile and prone to lodging on the medium, 2.375–10.710 mm high. The **sorophores** are colorless and slender, consisting of two to four tiers of cells, 6.539–17.767 μm wide. The **sorophore tips** are capitate, surrounded by a slimy substance, 5.539–8.011 μm in diameter. The **sorophore bases** are rounded, surrounded with a slimy substance, 16.922–39.985 μm in diameter. The **sori** is globose or lemon-shaped, hyaline or creamy white, transparent and hyaline in the early stage, and gradually tends to be milky white with time, 0.158–0.426 mm in diameter, mostly 0.284–0.313 mm in diameter. The **spores** are oblong or elliptical, elongated, 5.673–9.330 × 3.050–3.814 μm, without polar granules. The **aggregations** exhibit radiate streaming. The **pseudoplasmodia** migrate with the stalks.

Etymology: This name refers to the slender sorophores.

Holotype: HMJAU MR 430, HMJAU MR 431, and HMJAU MR 432 (strain 6709C, strain 6709L, and strain 6709M) were isolated from a soil sample collected in China, Jilin Province, Baishan City, Changbai Korean Autonomous County, Wangtian’e New Village (soil no. 6709; elevation 630 m; coordinates 41°27′44″ N, 127°56′31″ E), on 21 July 2021 in a scallion farmland.

The GenBank accession numbers for the SSU sequences are PQ287291, PQ287298, and PQ287299.

Known distribution: Currently, it is believed to be found only in China.

Commentary: The proliferation rate of this species is remarkable, with mature sorocarps emerging within a mere 48 h of cultivation. These sorocarps exhibit a distinctive feature—a notably elongated sorophore, which leads to the easy collapse of the sorocarps. Molecular phylogeny based on SSU rRNA supported the position of this species in the genus *Dictyostelium* [25] (Figure 9a). It forms a clade together with *D. brunneum* and *D. giganteum* [12,31]. 

However, the sorophore of *D. giganteum* is hyaline and sparsely branched, whereas *D. longigracilis* is unbranched. The spores of *D. longigracilis* (5.673–9.330 × 3.050–3.814 μm) are longer than those of *D. giganteum* (5.0–6.5 × 3.0–3.5 μm). Compared to *D. brunneum* (color of sori off-white to cream-colored when young, becoming tan to dull brown with age), the sori in *D. longigracilis* is colorless or milky white. Moreover, in terms of sorocarp size, it is worth noting the differences between *D. longigracilis* and another species, *D. longosporum*. The sorocarps of *D. longosporum* are 0.5–5.1 (−30) mm, whereas, in *D. longigracilis*, they are larger, ranging from 2.375 to 10.710 mm. This comparison highlights the variability in sorocarp size among different species within the genus.

***Dictyostelium macrosoriobrevipes***, found by Z.J. Zhang, P. Liu, and Y. Li sp. nov., is shown in Figure 3.

Fungal Names: FN 572068.

When cultured at 23 °C on non-nutrient agar with *Escherichia coli*, the **sorocarps** are mostly solitary, occasionally clustered, erect, semi-erect, or inclined, without branches or occasional branches, 0.483–1.389 mm high. The **sorophores** are thick and short, consisting of two to five tiers of cells, tapering from the base to the top, 18.653–27.520 μm wide. The **sorophore tips** are obtuse, with a collar structure, with two to three tiers of cells, relatively thick, 10.616–19.303 μm in diameter. The **sorophore bases** are clavate, 14.860–22.027 μm in diameter. The **sori** is globose, hyaline or milky white, 0.065–0.578 mm in diameter, but mostly 0.216–0.348 mm in diameter. The **spores** are oblong or reniform, 9.282–17.484 × 4.527–6.382 μm, without polar granules. The **aggregations** exhibit radiate streaming, and the sorocarps are pulled up from the center of the aggregations or the end of the aggregations. The **sorogens** are colorless, 0.416–1.619 mm in length. The **pseudoplasmodia** migrate with the stalks.

Etymology: This name refers to the large sori and the short sorophores.

Holotype: HMJAU MR433 (Strain 6718S1-2B) was isolated from a soil sample from China, Jilin Province, Baishan City, Changbai Korean Autonomous County, Primary Protection Zone of Drinking Water Sources (soil no. 6718; elevation 717.85 m; coordinates 41°29′17″ N, 127°56′32″ E), collected on 23 July 2021, in a broadleaf forest.

The GenBank accession number for the SSU sequence is PQ304778.

Known distribution: Currently, it is believed to be found only in China.

Commentary: *Dictyostelium macrosoriobrevipes* is characterized by a collar-like structure at the tip of the sorocarps, large sori, and a short sorophore compared to its large sori. A molecular phylogeny study based on SSU rRNA supported the position of this species in the genus *Dictyostelium* [25] (Figure 9a). This species forms a clade with *D. robusticaule* [21] and *D. minimum* [42]. However, the sorocarps of *D. robusticaule* are 0.3–2.4 mm or 0.1–0.2 mm, whereas those of *D. macrosoriobrevipes* are 0.483–1.389 mm. Radiate aggregations occur in *D. macrosoriobrevipes*, whereas those of *D. minimum* are mounds, and *D. multiforme* aggregations are mound-like and minutum-type. The spores of *D. robusticaule* (5.0–7.2 × 3.2–4.7 μm or 4.7–6.5 × 3.0–4.3 μm) and *D. minimum* (3.7–5.4 μm) are smaller than in *D. macrosoriobrevipes* (9.282–17.484 × 4.527–6.382 μm).

***Dictyostelium robusticaule***, found by Y. Li, P. Liu, and Y. Zou and described by Zou, Hou, Guo, Li, Li, Stephenson, Pavlov, Liu and Lia in *Microbiology Spectrum* 10 (5): 8 (2022), is shown in Figure 4.

When cultured at 23 °C on non-nutrient agar with *Escherichia coli*, the **sorocarps** are erect or twisted in the medium, with branches or with branches which are not obvious in the early stage; sometimes, branches will appear in the later stage. They are 1.601–3.182 mm high. The **sorophores** are colorless, consisting of three to four tiers of cells, 19.235–21.152 μm wide. The **sorophore bases** are clavate, while the **sorophore tips** have a collar structure. The **sori** is large, globose, hyaline or milk white, 0.126–0.432 mm in diameter. The **spores** are oblong or elliptical, 7.156–9.612 × 4.449–6.249 μm, without polar granules. The **aggregations** have a radiate pattern, and **pseudoplasmodia** migration occurs with the sorophore.

Holotype: HMJAU MR 434 (strain 6700D) was isolated from a soil sample from China, Jilin Province, Baishan City, Changbai Korean Autonomous County, Fifteen Gap (soil no. 6700; elevation 651 m; coordinates 41°28′03″ N, 127°56′26″ E), collected on July 20, 2021, in a broadleaf forest. 

The GenBank accession number for the SSU sequence is PQ237105.

Known distribution: Currently, it is believed to only be found in China, specifically in Guizhou [22] and Jilin [21].

***Polysphondylium patagonicum,*** described by Vadell, Cavender, Romeralo, and S.L. Stephenson in *Mycologia* 103 (1): 113 (2011), is shown in Figure 5.

When cultured at 23 °C on non-nutrient agar with *Escherichia coli*, the **sorocarps** are solitary or gregarious, erect or semi-erect, often prostrate on the medium, unbranched or with 2–8 whorled branches, each whorl with 2–5 spaced nodes, nearly 2.85–6.32 mm high. The **sorophores** are purplish red, consisting of several tiers of cells. The **sorophore tips** are clavate, 6.45–11.79 μm in diameter. The **sorophore bases** are round or clavate, 10.14–21.99 μm in diameter. The **sori** is globose, light purple or purplish red, 0.075–0.233 mm in diameter. The **spores** are elliptical, mostly 5.47–6.75 × 3.084–4.314 μm, with consolidated polar granules. The **aggregations** have a radiate pattern. The **pseudoplasmodia** migrate with the stalks.

Holotype: HMJAU MR 435 and HMJAU MR 436 (strain 6718-Z2, strain 6718-zi) were isolated from a soil sample of China, Jilin Province, Baishan City, Changbai Korean Autonomous County, Primary Protection Zone of Drinking Water Sources (soil no. 6718; elevation 717.85 m; coordinates 41°29′17″ N, 127°56′32″ E), collected on 23 July 2021 in a broadleaf forest. 

The GenBank accession numbers for the SSU sequences are PQ236756 and PQ236757.

Known distribution: It can be found in Patagonia [43], Argentina, Russia [44], and China, specifically in Jilin.

***Heterostelium candidum*** (H. Hagiw.), described by S. Baldauf, S. Sheikh, and Thulin, in Sheikh, Thulin, Cavender, Escalante, Kawakami, Lado, Landolt, Nanjundiah, Queller, Strassmann, Spiegel, Stephenson, Vadell, and Baldauf, *Protist* 169: 14 (2018), is shown in Figure 6.

When cultured at 23 °C on non-nutrient agar with *Escherichia coli*, the **sorocarps** are solitary, sometimes clustered, inclined or prostrate on the medium, 1.367–5.574 mm high. The **sorophores** are colorless, consisting of one to two tiers of cells, with about 3–7 whorled branches, each whorl bearing 2–4 sori. The **sorophore bases** are clavate, their branch end conical. The **sori** is globose, white, and transparent; the terminal sori measures 65.27–100.92 μm in diameter, while the lateral sori is 34.19–82.27 μm in diameter. The **spores** are elliptical to oblong, 7.116–10.426 × 3.266–5.959 μm, with polar granules. The **aggregations** have a radiate pattern. The **pseudoplasmodia** migrate with the stalks.

Holotype: HMJAU MR 437 (strain L1-1B) was isolated from soil samples collected in China, Jilin Province, Baishan City, Changbai Korean Autonomous County, Lishui Manor (soil no. 6715; elevation 795 m; coordinates 41°31′23″ N, 127°56′58″ E), on 22 July 2021 in a broadleaf forest. 

The GenBank accession number for the SSU sequence is PQ222844.

Known distribution: It can be found in Japan, United States, and China, specifically in Hubei, Henan [45], and Jilin. 

***Cavenderia aureostipes*** (Cavender), found by S. Baldauf, S. Sheikh and Thulin and described in Sheikh, Thulin, Cavender, Escalante, Kawakami, Lado, Landolt, Nanjundiah, Queller, Strassmann, Spiegel, Stephenson, Vadell, and Baldauf, *Protist* 169: 19 (2018), is shown in Figure 7.

When cultured at 23 °C on non-nutrient agar with *Escherichia coli*, the **sorocarps** are solitary or clustered, erect or semi-erect, with irregular branches, strongly phototropic, and nearly 3.81–7.53 mm high. The **sorophores** are golden yellow, with 5–23 irregular branches, 392.52–749.81 μm wide. The **sorophore tips** are obtuse, 7.48–10.82 μm in diameter. The **sorophore bases** are clavate, consisting of several tiers of cells, 23.26–27.79 μm in diameter. The **sori** is white and globose; the terminal sori is 140.71–194.48 μm in diameter, while the lateral sori measures 36.14.78–82.81 μm in diameter. The **spores** are elliptical to oblong, mostly 4.733–6.281 × 2.881–4.251 μm, with consolidated polar granules. The **aggregations** have a radiate pattern. The **pseudoplasmodia** migrate with the stalks.

Holotype: HMJAU MR 438 (strain 6709-2) was isolated from a soil sample collected in China, Jilin Province, Baishan City, Changbai Korean Autonomous County, Wangtian’e New Village (soil no. 6709; elevation 630 m; coordinates 41°27′44″ N, 127°56′31″ E), on 21 July 2021, on scallion farmland. HMJAU MR 439 (strain L1-2L) was isolated from soil samples from China, Jilin Province, Baishan City, Changbai Korean Autonomous County, Lishui Manor (soil no. 6715; elevation 795 m; coordinates 41°31′23″ N, 127°56′58″ E), collected on 22 July 2021, in a broadleaf forest. HMJAU MR 440, HMJAU MR 441, and HMJAU MR 442 (strain L2-1A, strain L2-113A, and strain L2-113B) were isolated from soil samples collected in China, Jilin Province, Baishan City, Changbai Korean Autonomous County, Lishui Manor (soil no. 6716; elevation 798.47 m; coordinates 41°31′22″ N, 127°57′0″ E), on 22 July 2021, in a broadleaf forest. HMJAU MR 443 and HMJAU MR 444 (strain S1-3A and strain S2-2A) were isolated from soil samples from China, Jilin Province, Baishan City, Changbai Korean Autonomous County, Primary Protection Zone of Drinking Water Sources (soil no. 6718; elevation 717.85 m; coordinates 41°29′17″ N, 127°56′32″ E; and soil no. 6719; elevation 729.46 m; coordinates 41°29′17″ N, 127°56′34″ E), collected on 23 July 2021, in broadleaf forests.

The GenBank accession numbers for the SSU sequences are PQ222850, PQ222849, PQ222848, PQ222847, PQ222846, PQ222845, and PQ222842.

Known distribution: It is found in Switzerland, the United States, Japan, India, Nepal, Russia, and China, specifically on the Qinghai–Tibet Plateau [42] and in Jilin. 

### 3.2. Assemblage of Dictyostelids in Jilin Province

Based on the identification, collation, and statistical analysis carried out in this study, a total of 35 species of dictyostelids, classified under 6 genera, 4 families, and 2 orders, are known to occur in Jilin Province (Appendix A, Figure 12a). Their primary ecological niche is soil, with a minority occurring in decaying leaves and humus (Appendix A). The dominant order within this group is the Dictyosteliales (Figure 12b), featuring 1 predominant family (>10 species) (Figure 12c), encompassing 20 species, accounting for 57% of the total species count. Within the group in Jilin Province, there are 2 dominant genera (>5 species) (Figure 12d), comprising 25 species, dominated by *Dictyostelium* and *Polysphondylium*, which, together, represent 71% of the total species diversity.

## 4. Discussion

### 4.1. Species Distribution and Geographic Uniqueness

As a hotspot for ecological diversity conservation in Northeast China, Changbai Korean Autonomous County [46] (Figure 1a) in Jilin Province boasts a unique geographical location (nestled in the heart of Changbai Mountain, featuring a mild and humid climate with four distinct seasons) and a complex terrain (encompassing various ecosystems such as mountains, forests, and wetlands), providing an ideal habitat for microorganisms like dictyostelids.

In this study, we collected 28 soil samples from five localities in Changbai Korean Autonomous County, Jilin Province (Table 1), and employed a combined approach of morphological [12,31] (Figure 2, Figure 3, Figure 4, Figure 5, Figure 6 and Figure 7) and the latest molecular systematic [25] (Figure 8, Figure 9, Figure 10 and Figure 11) identification methods, yielding 15 dictyostelid isolates belonging to 6 species from 4 genera, with the family Dictyosteliaceae occupying a dominant position (66.67%). The family Dictyosteliaceae [47], the most species-rich group within the class Dictyosteliales, primarily comprises the genera *Dictyostelium* and *Polysphondylium* [25]. Due to the special geographical location and terrain of Jilin Province, it is characterized by some unique species of dictyostelids, including two new species and two new records, not only reflecting the abundance of dictyostelid resources in this region but also underlining its significance in microbial diversity conservation.

### 4.2. Ecological Significance and Species Diversity

The terrain of Changbai Korean Autonomous County is dominated by mountains and hills, with elevations ranging from 500 to 2691 m. The total forest area of Changbai Korean Autonomous County is 232,895.6 hectares, representing a forest coverage of 92%, ranking among the forefront in the province and even the country (http://changbai.gov.cn/cbgk/zrzy/201801/t20180108_257152.html, accessed on 11 May 2023). Dictyostelids, as a crucial component of soil ecosystems, engage in processes such as organic matter decomposition and nutrient cycling, thereby playing a pivotal role in maintaining ecological balance and soil health [22,48]. 

The five localities investigated encompassed four vegetation types: coniferous forest, broadleaf forest, mixed forest, and farmland. Overall, the density and fruiting body abundance of dictyostelids in broadleaf forests was up to 19.44 clones/g, followed by farmland with 1.67 clones/g. No isolates were recovered from the coniferous forest at L1 in this study, and no isolates were obtained from any of the sampled mixed forests. Previous research by Zou et al. [21] and Zhang et al. [11] indicated that mixed forests yielded more isolates, whereas our findings revealed the highest species richness in broadleaf forests. As is generally known, the growth and reproduction of soil protists depend largely on the habitat and substrate provided by plants [49]. However, the composition and quantity of protozoa communities vary significantly under different vegetation types [50], highlighting the notable impact of vegetation type on the richness of dictyostelids at sampling sites.

### 4.3. Composition and Distribution of Dictyostelid Species in Jilin Province

Furthermore, this study primarily relied on comprehensive literature sources such as mycological treatises [51] and the published literature to provide a detailed inventory of dictyostelid distributions at the county and city levels. Corresponding voucher information is also provided for each species’ occurrence in Jilin Province, encompassing specimens, substrate numbers, vegetation types, strain numbers, geographical locations, and references. These voucher details serve as verifiable evidence for researchers to examine. 

Notably, in 1984, Bai [52] first reported five species of dictyostelids from Jilin Province. To date, Jilin Province is known to have 35 species, belonging to 6 genera, 4 families, and 2 orders (Appendix A). Among them, the dominant family is the Dictyosteliaceae, and the dominant genus is *Dictyostelium* (Figure 12). Swanson et al. [53] divided the global distribution of 65 species of forest soil dictyostelids into 4 categories—cosmopolitan, disjunct, restricted, and pantropical. The dictyostelid species in Jilin Province are abundant, and, in terms of the geographical composition of families and genera, they are dominated by restricted species, forming a complex and diverse assemblage. This is also consistent with the distribution of Madagascar species reported by Cavender et al. [54], indicating that species distribution is sometimes regional.

## 5. Conclusions

The present study underscores the significance of dictyostelids within the protist community, particularly in Jilin Province, China. Through a comprehensive resource survey employing random sampling at five localities in Changbai Korean Autonomous County, we identified a rich diversity of dictyostelids, including the description of two novel species based on morphological and genetic analyses. The dominance of the genus *Dictyostelium* in the dictyostelid assemblage of Jilin Province, coupled with the discovery of new records for China and Jilin Province, highlights the region’s importance as a hotspot for these organisms. Our findings not only contribute to the understanding of dictyostelid diversity and distribution but also provide valuable insights into their ecological roles and potential applications. The compilation of dictyostelid species reported in Jilin Province represents an important step towards developing effective conservation strategies and promoting sustainable resource utilization of these fascinating organisms.

## Figures and Tables

**Figure 1 jof-10-00834-f001:**
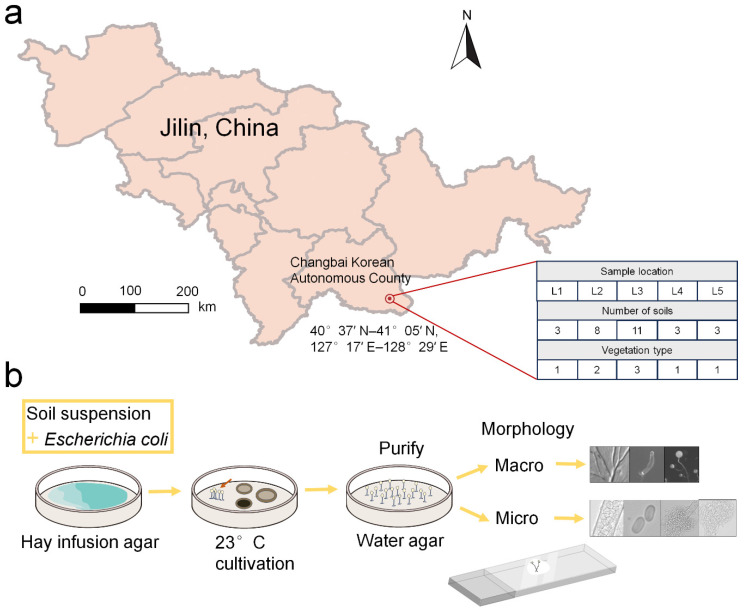
Study area and location of the sampling sites in Changbai Korean Autonomous County, Jilin Province, China (**a**). Outline of the experimental workflow used to isolate dictyostelids from the collected soil samples (**b**), the arrow indicates the step-by-step progression of the experiment. Note: the map was drawn using ArcGIS 10.8 [28] software. Details of the samples (e.g., sample location, number of soil samples, and vegetation types) are shown in Table 1.

**Figure 2 jof-10-00834-f002:**
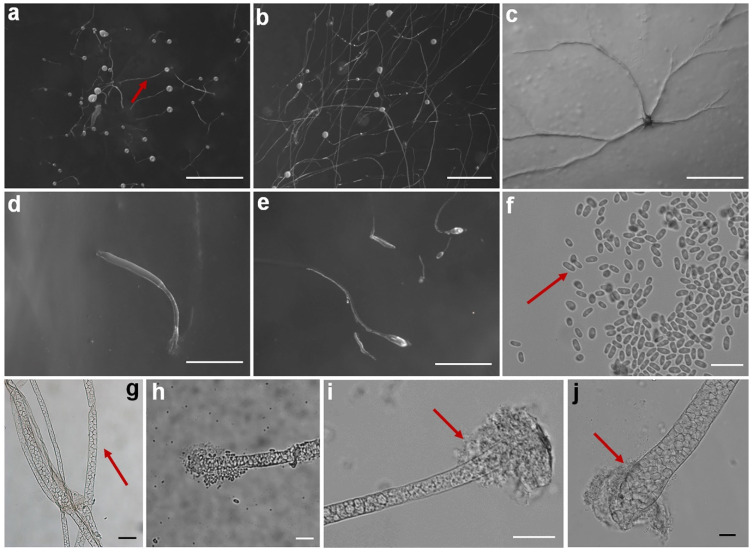
Morphological features of *Dictyostelium longigracilis*: (**a**,**b**) sorocarps; (**c**) aggregation; (**d**,**e**) pseudoplasmodia; (**f**) spores; (**g**) sorophores; (**h**,**i**) sorophore tips; and (**j**) sorophore base. Scale bars: (**a**–**c**) = 2 mm; (**d**,**e**) = 1 mm; (**f**,**i**,**j**) = 20 μm; (**g**) = 50 μm; and (**h**) = 10 μm. The red arrows refer to the key distinguishing characteristics.

**Figure 3 jof-10-00834-f003:**
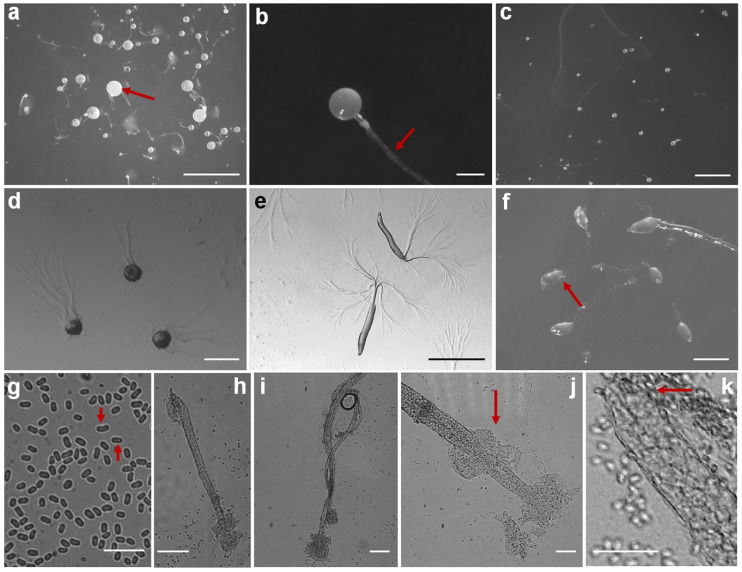
Morphological features of *Dictyostelium macrosoriobrevipes*: (**a**–**c**) sorocarps; (**d**) aggregations; (**e**) pseudoplasmodia; (**f**) sorogens; (**g**) spores; (**h**,**i**) sorophores; (**j**) tip of sorophore; and (**k**) base of sorophore. Scale bars: (**a**,**c**) = 2 mm; (**b**) = 200 μm; (**d**,**f**) = 500 μm; (**e**) = 1 mm; (**g**–**j**) = 50 μm; and (**k**) = 20 μm. The red arrows refer to the key distinguishing characteristics.

**Figure 4 jof-10-00834-f004:**
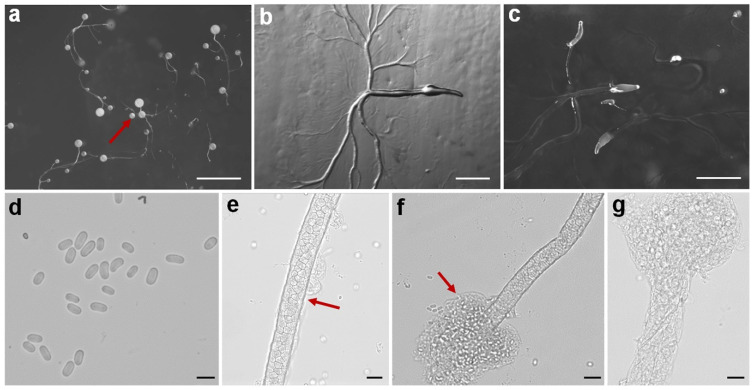
Morphological features of *Dictyostelium robusticaule*: (**a**) sorocarps; (**b**) aggregations; (**c**) pseudoplasmodia; (**d**) spores; (**e**) sorophore; (**f**) sorophore tip; and (**g**) sorophore base. Scale bars: (**a**) = 2 mm; (**b**,**c**) = 1 mm; and (**d**–**g**) = 20 μm. The red arrows refer to the key distinguishing characteristics.

**Figure 5 jof-10-00834-f005:**
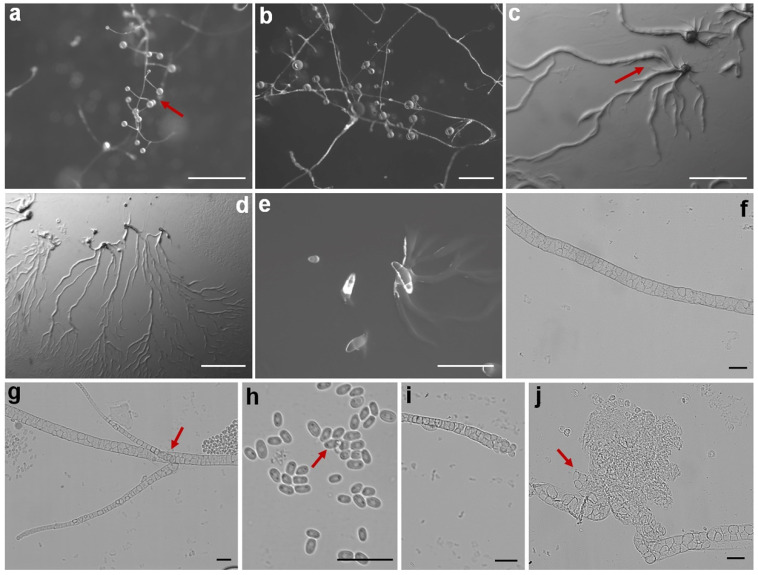
Morphological features of *Polysphondylium patagonicum*: (**a**,**b**) sorocarps; (**c**,**d**) aggregations; (**e**) pseudoplasmodia; (**f**) sorophore; (**g**) branches; (**h**) spores; (**i**) sorophore tip; and (**j**) sorophore base. Scale bars: (**a**,**c**) = 1 mm; (**b**,**e**) = 500 μm; (**d**) =2 mm; and (**f**–**j**) = 20 μm. The red arrows refer to the key distinguishing characteristics.

**Figure 6 jof-10-00834-f006:**
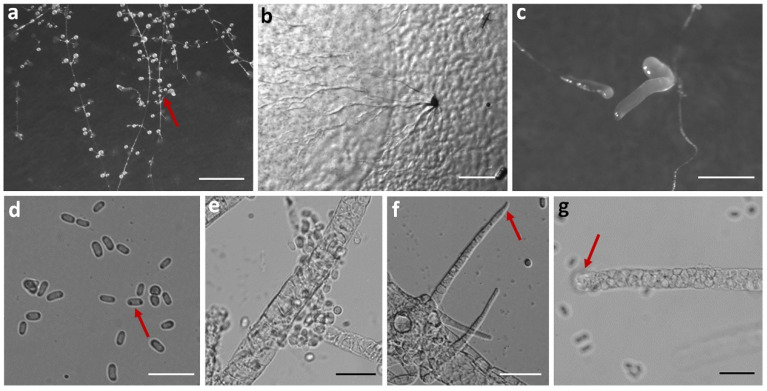
Morphological features of *Heterostelium candidum*: (**a**) sorocarps; (**b**) aggregations; (**c**) pseudoplasmodia; (**d**) spores; (**e**) sorophore; (**f**) sorophore tip; and (**g**) sorophore base. Scale bars: (**a**,**b**) = 1 mm; (**c**) = 500 μm; (**d**) = 20 μm; (**e**) = 10 μm; (**f**) = 50 μm; and (**g**) = 20 μm. The red arrows refer to the key distinguishing characteristics.

**Figure 7 jof-10-00834-f007:**
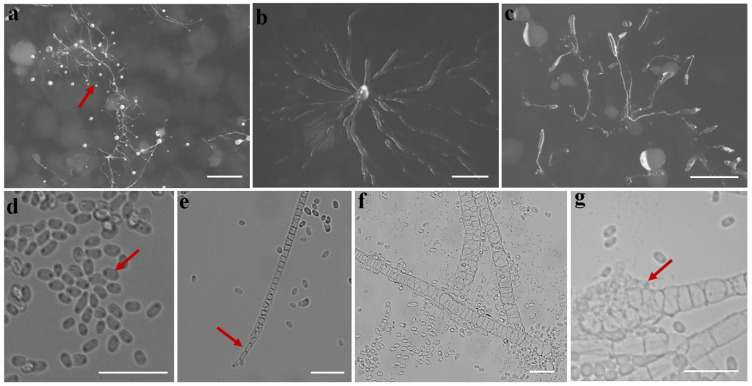
Morphological features of *Cavenderia aureostipes*: (**a**) sorocarps; (**b**) aggregations; (**c**) pseudoplasmodia; (**d**) spores; (**e**) sorophore tip; (**f**) sorophore; and (**g**) sorophore base. Scale bars: (**a**,**c**) = 1 mm; (**b**) = 500 μm; and (**d**–**g**) = 20 μm. The red arrows refer to the key distinguishing characteristics.

**Figure 8 jof-10-00834-f008:**
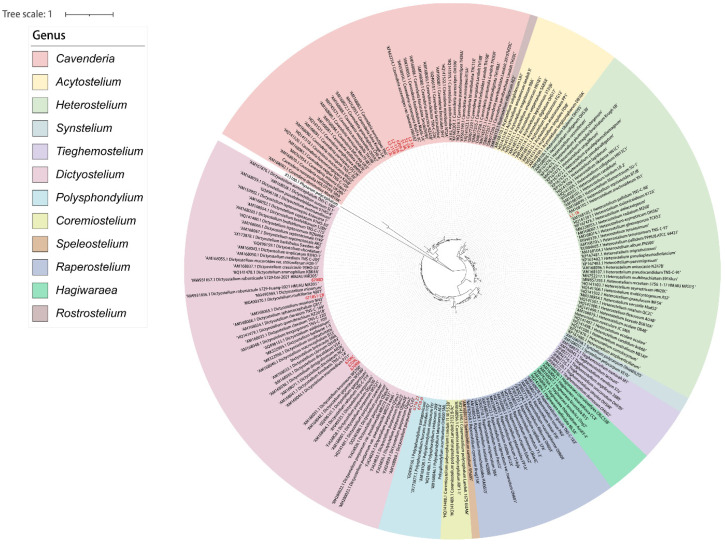
Phylogeny of all known species of dictyostelids based on SSU rDNA. Newly generated sequences are indicated in red.

**Figure 9 jof-10-00834-f009:**
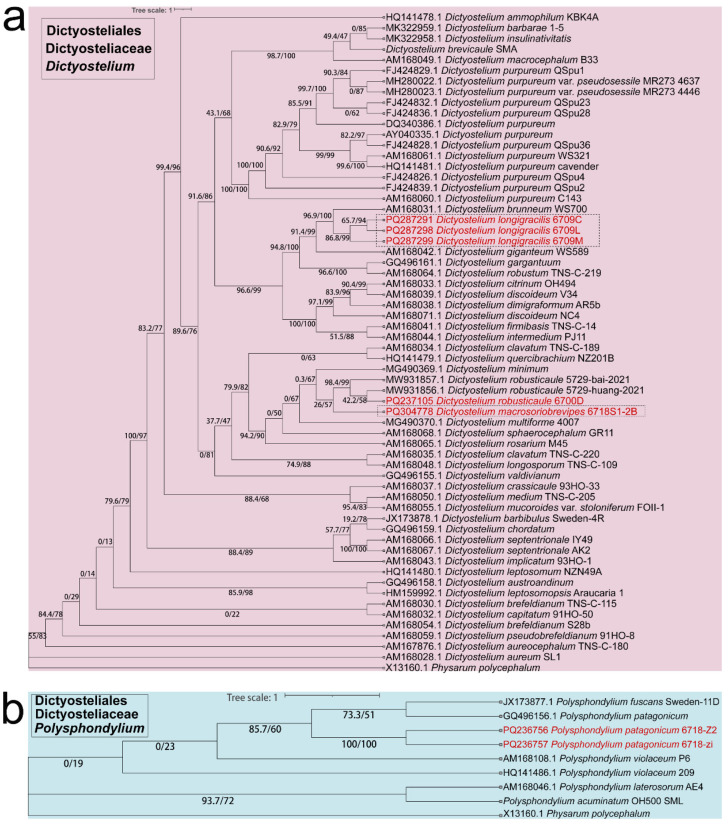
SSU phylogeny of *Dictyostelium* (**a**) and *Polysphondylium* (**b**) sequences in the order Dictyosteliales and the family Dictyosteliaceae. Numbers in parentheses are SH-aLRT support (%)/ultrafast bootstrap support (%). Newly generated sequences are indicated in red and the new species are framed in black.

**Figure 10 jof-10-00834-f010:**
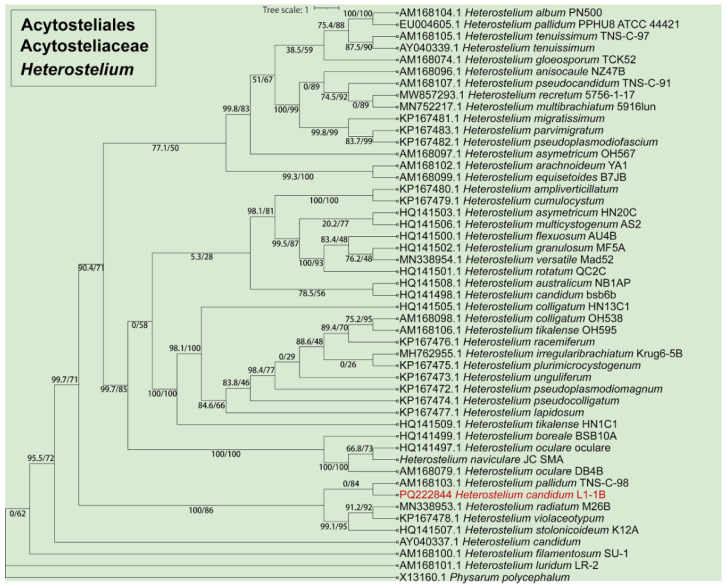
SSU phylogeny of *Heterostelium* sequences in the order Acytosteliales and the family Acytosteliaceae. Numbers in parentheses are SH-aLRT support (%)/ultrafast bootstrap support (%). Newly generated sequences are indicated in red.

**Figure 11 jof-10-00834-f011:**
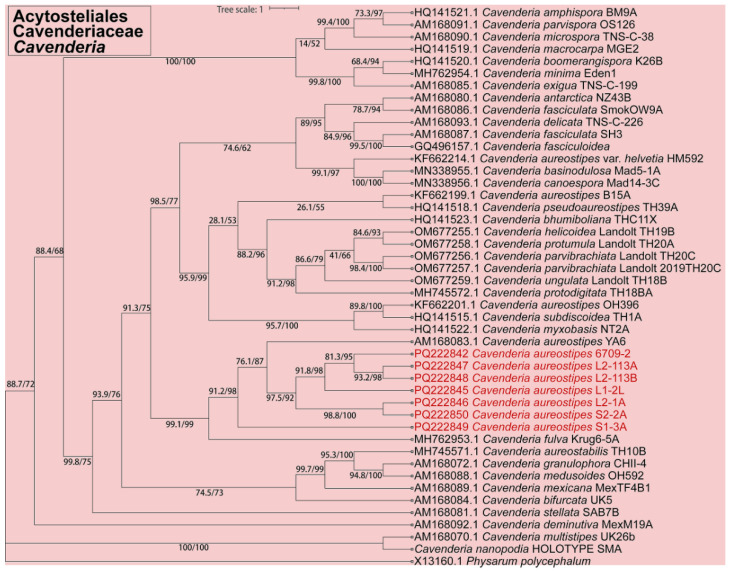
SSU phylogeny of *Cavenderia* sequences in the order Acytosteliales and the family Cavenderiaceae. Numbers in parentheses are SH-aLRT support (%)/ultrafast bootstrap support (%). Newly generated sequences are indicated in red.

**Figure 12 jof-10-00834-f012:**
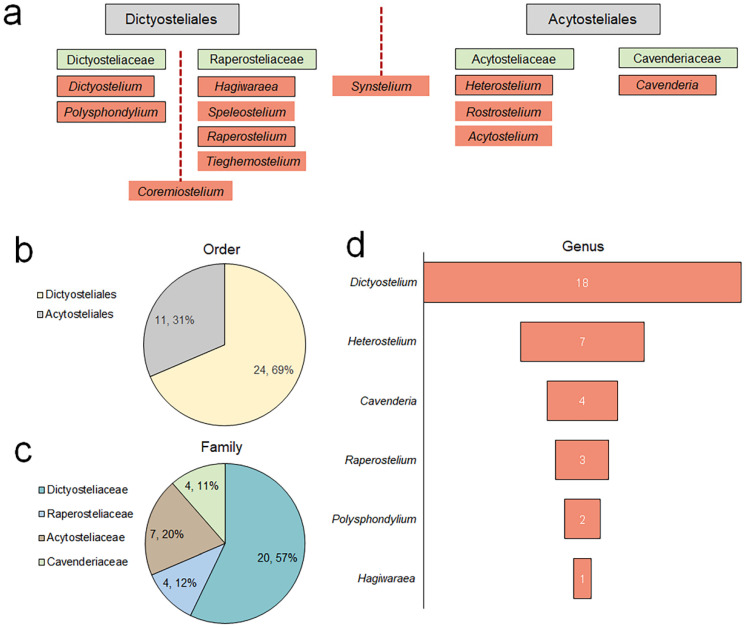
The new classification as indicated by nuclear small subunit (18S) [25] rDNA sequences, displayed with a black border (**a**) and numbers of dictyostelid species within the order (**b**), family (**c**), and genus (**d**) in Jilin Province, China.

**Table 1 jof-10-00834-t001:** Sample location information and dictyostelid species isolated from Changbai Korean Autonomous County, Jilin Province.

Soil Nos.	Localities	Vegetation	Coordinates	Elevation (m)	Temperature	Dictyostelid Species
6693	Tashan Park [L1]	Coniferous forest	41°25′27″ N, 128°11′36″ E	833	19–27 °C	——
6694			41°25′29″ N, 128°11′36″ E	854.62	19–27 °C	——
6695			41°25′3″ N, 128°11′35″ E	849	19–27 °C	——
6696	Fifteen Gap [L2]	Farmland (soybean field)	41°28′1″ N, 127°56′28″ E	655	19–27 °C	——
6697			41°28′02″ N, 127°56′27″ E	652	19–27 °C	——
6698			41°28′02″ N, 127°56′27″ E	655	19–27 °C	——
6699		Broadleaf forest	41°27′02″ N, 127°56′18″ E	644	19–27 °C	——
6700			41°28′03″ N, 127°56′26″ E	651	19–27 °C	*** *Dictyostelium longigracilis* (3), *Dictyostelium* *robusticaule* (1)
6701			41°27′17″ N, 127°56′26″ E	648	19–27 °C	——
6702			41°27′17″ N, 127°56′13″ E	644	19–27 °C	——
6703			47°28′0″ N, 127°56′26″ E	655.64	19–27 °C	——
6704	Wangtian’e New Village [L3]	Mixed forest	41°27′27″ N, 127°56′41″ E	682	19–30 °C	——
6705			41°27′26″ N, 127°56′41″ E	686	19–30 °C	——
6706			41°22′25″ N, 127°56′42″ E	732.13	19–30 °C	——
6707		Broadleaf forest	41°27′27″ N, 127°56′41″ E	678	19–30 °C	——
6708		Farmland (scallion field)	41°27′41″ N, 127°56′32″ E	654	19–30 °C	——
6709		Farmland (scallion field)	41°27′44″ N, 127°56′31″ E	630	19–30 °C	* *Cavenderia* *aureostipes* (1)
6710		Farmland (corn field)	41°27′40″ N, 127°56′32″ E	680	19–30 °C	——
6711		Farmland (corn field)	41°27′41″ N, 127°56′32″ E	671	19–30 °C	——
6712		Farmland (corn field)	41°27′41″ N, 127°56′33″ E	675	19–30 °C	——
6713		Farmland (Chinese chives field)	41°27′44″ N, 127°56′31″ E	675	19–30 °C	——
6714		Farmland (soybean field)	41°27′42″ N, 127°56′32″ E	652	19–30 °C	——
6715	Lishui Manor [L4]	Broadleaf forest	41°31′23″ N, 127°56′58″ E	795	19–30 °C	*Heterostelium* *candidum* (1), * *Cavenderia* *aureostipes* (1)
6716			41°31′22″ N, 127°57′0″ E	798.47	19–30 °C	* *Cavenderia* *aureostipes* (3)
6717			41°31′24″ N, 127°57′1″ E	787	19–30 °C	——
6718	Primary Protection Zone of Drinking Water Sources [L5]	Broadleaf forest	41°29′17″ N, 127°56′32″ E	717.85	19–30 °C	* *Cavenderia* *aureostipes* (1), ** *Polysphondylium* *patagonicum* (2), *** *Dictyostelium macrosoriobrevipes* (1)
6719		Broadleaf forest	41°29′17″ N, 127°56′34″ E	729.46	19–30 °C	* *Cavenderia* *aureostipes* (1)
6720		Broadleaf forest	41°29′16″ N, 127°56′34″ E	693.39	19–30 °C	——

Numbers (1), (2), and (3) indicate the number of species clones/strains obtained from the soil samples (n). * refers to a species new to Jilin Province. ** refers to a species new to China. *** refers to a species new to science.

**Table 2 jof-10-00834-t002:** Sequences used in phylogenetic analysis based on SSU.

Species Name	Strain Number	SSU Accession Numbers	Species Name	Strain Number	SSU Accession Numbers
*Cavenderia amphispora*	BM9A	HQ141521.1	*T. lacteum*		AM168045.1
*C. antarctica*	NZ43B	AM168080.1	*T. menorah*	M1	AM168073.1
*C. aureostabilis*	TH10B	MH745571.1	*T. montium*	57a	JF892717.1
*C. aureostipes*	YA6	AM168083.1	*T. simplex*	OH598	JF892720.1
*C. aureostipes*	B15A	KF662199.1	*T. unicornutum*	OH599	JF892725.1
*C. aureostipes*	OH396	KF662201.1	*Hagiwaraea coeruleostipes*	CRLC53B	AM168036.1
** *C. aureostipes* **	**S2-2A**	**PQ222850**	*H. lavandula*	B15	AM168047.1
** *C. aureostipes* **	**S1-3A**	**PQ222849**	*H. radiculata*	ML5A	HQ141494.1
** *C. aureostipes* **	**L2-113B**	**PQ222848**	*H. rhizopodium*	AusKY-4	AM168063.1
** *C. aureostipes* **	**L2-113A**	**PQ222847**	*H. tenebrica*	Ong2	MH762956.1
** *C. aureostipes* **	**L2-1A**	**PQ222846**	*H. vinaceofusca*	CC4	AM168062.1
** *C. aureostipes* **	**L1-2L**	**PQ222845**	*Raperostelium australe*	NZ80B	AM168029.1
** *C. aureostipes* **	**6709-2**	**PQ222842**	*R. capillare*	37A	JF892721.1
*C. aureostipes* var. *helvetia*	HM592	KF662214.1	*R. crispum*	Eden2	MH762957.1
*C. basinodulosa*	Mad5-1A	MN338955.1	*R. cymosum*	Krug15A	MH762958.1
*C. bhumiboliana*	THC11X	HQ141523.1	*R. filiforme*	OH603	JF892724.1
*C. bifurcata*	UK5	AM168084.1	*R. gracile*	TNS-C-183	AM168078.1
*C. boomerangispora*	K26B	HQ141520.1	*R. ibericum*	214rjb	HQ141495.1
*C. canoespora*	Mad14-3C	MN338956.1	*R. maeandriforme*	OH604	JF892719.1
*C. delicata*	TNS-C-226	AM168093.1	*R. minutum*	71-2	AM168051.1
*C. deminutiva*	MexM19A	AM168092.1	*R. monochasioides*	HAG653	AM168052.1
*C. exigua*	TNS-C-199	AM168085.1	*R. ohioense*	Okla4C	HQ141493.1
*C. fasciculata*	SmokOW9A	AM168086.1	*R. potamoides*	FP1A	AM168069.1
*C. fasciculata*	SH3	AM168087.1	*R. reciprocatum*	38A	JF892718.1
*C. fasciculoidea*		GQ496157.1	*R. reciprocatum* var. *transitum*	OH601	JF892723.1
*C. fulva*	Krug6-5A	MH762953.1	*R. stabile*	M12A	MN338957.1
*C. granulophora*	CHII-4	AM168072.1	*R. tenue*	Pan52	AM168076.1
*C. helicoidea*	Landolt TH19B	OM677255.1	*R. tenue*	PJ6	AM168094.1
*C. macrocarpa*	MGE2	HQ141519.1	*R. tenue*	PR4	AM168075.1
*C. medusoides*	OH592	AM168088.1	*Speleostelium caveatum*	WS695	AM168077.1
*C. mexicana*	MexTF4B1	AM168089.1	*Dictyostelium ammophilum*	KBK4A	HQ141478.1
*C. microspora*	TNS-C-38	AM168090.1	*D. aureocephalum*	TNS-C-180	AM167876.1
*C. minima*	Eden1	MH762954.1	*D. aureum*	SL1	AM168028.1
*C. multistipes*	UK26b	AM168070.1	*D. austroandinum*		GQ496158.1
*C. myxobasis*	NT2A	HQ141522.1	*D. barbarae*	1-5	MK322959.1
*C. parvibrachiata*	Landolt TH20C	OM677256.1	*D. barbibulus*	Sweden-4R	JX173878.1
*C. parvibrachiata*	Landolt 2019 TH20C	OM677257.1	*D. brefeldianum*	TNS-C-115	AM168030.1
*C. parvispora*	OS126	AM168091.1	*D. brunneum*	WS700	AM168031.1
*C. protodigitata*	TH18BA	MH745572.1	*D. capitatum*	91HO-50	AM168032.1
*C. protumula*	Landolt TH20A	OM677258.1	*D. chordatum*		GQ496159.1
*C. pseudoaureostipes*	TH39A	HQ141518.1	*D. citrinum*	OH494	AM168033.1
*C. stellata*	SAB7B	AM168081.1	*D. clavatum*	TNS-C-189	AM168034.1
*C. subdiscoidea*	TH1A	HQ141515.1	*D. clavatum*	TNS-C-220	AM168035.1
*C. ungulata*	Landolt TH18B	OM677259.1	*D. crassicaule*	93HO-33	AM168037.1
*Acytostelium amazonicum*	Landolt X	HQ141510.1	*D. dimigraformum*	AR5b	AM168038.1
*A. amazonicum*	HN1B1	HQ141511.1	*D. discoideum*	NC4	AM168071.1
*A. anastomosans*	PP1	AM168115.1	*D. discoideum*	V34	AM168039.1
*A. digitatum*	OH517	AM168114.1	*D. firmibasis*	TNS-C-14	AM168041.1
*A. leptosomum*	212rjb	HQ141512.1	*D. gargantuum*		GQ496161.1
*A. leptosomum*	FG12	AM168111.1	*D. giganteum*	WS589	AM168042.1
*A. longisorophorum*	DB10A	AM168109.1	** *D. longigracilis* **	**6709C**	**PQ287291**
*A. magnisorum*	08A	HQ141513.1	** *D. longigracilis* **	**6709L**	**PQ287298**
*A. serpentarium*	SAB3A	AM168113.1	** *D. longigracilis* **	**6709M**	**PQ287299**
*A. singulare*	FDIB	HQ141514.1	*D. implicatum*	93HO-1	AM168043.1
*A. subglobosum*	LB1	AM168110.1	*D. insulinativitatis*		MK322958.1
*Rostrostelium ellipticum*	AE2	AM168112.1	*D. intermedium*	PJ11	AM168044.1
*Heterostelium album*	PN500	AM168104.1	*D. leptosomopsis*	Araucaria 1	HM159992.1
*H. ampliverticillatum*		KP167480.1	*D. leptosomum*	NZN49A	HQ141480.1
*H. anisocaule*	NZ47B	AM168096.1	*D. longosporum*	TNS-C-109	AM168048.1
*H. arachnoideum*	YA1	AM168102.1	*D. macrocephalum*	B33	AM168049.1
*H. asymetricum*	OH567	AM168097.1	*D. medium*	TNS-C-205	AM168050.1
*H. asymetricum*	HN20C	HQ141503.1	*D. minimum*	2794	MG490369.1
*H. australicum*	NB1AP	HQ141508.1	*D. brefeldianum*	S28b	AM168054.1
*H. boreale*	BSB10A	HQ141499.1	*D. mucoroides* var. *stoloniferum*	FOII-1	AM168055.1
*H. candidum*	bsb6b	HQ141498.1	*D. multiforme*	4007	MG490370.1
*H. candidum*		AY040337.1	*D. pseudobrefeldianum*	91HO-8	AM168059.1
** *H. candidum* **	**L1-1B**	**PQ222844**	*D. purpureum*	QSpu1	FJ424829.1
*H. colligatum*	HN13C1	HQ141505.1	*D. purpureum*	QSpu2	FJ424839.1
*H. colligatum*	OH538	AM168098.1	*D. purpureum*	QSpu23	FJ424832.1
*H. cumulocystum*		KP167479.1	*D. purpureum*	QSpu28	FJ424836.1
*H. equisetoides*	B7JB	AM168099.1	*D. purpureum*		DQ340386.1
*H. filamentosum*	SU-1	AM168100.1	*D. purpureum*	QSpu4	FJ424826.1
*H. flexuosum*	AU4B	HQ141500.1	*D. purpureum*		AY040335.1
*H. gloeosporum*	TCK52	AM168074.1	*D. purpureum*	QSpu36	FJ424828.1
*H. granulosum*	MF5A	HQ141502.1	*D. purpureum*	C143	AM168060.1
*H. irregularibrachiatum*	Krug6-5B	MH762955.1	*D. purpureum*	WS321	AM168061.1
*H. lapidosum*		KP167477.1	*D. purpureum*	cavender	HQ141481.1
*H. luridum*	LR-2	AM168101.1	*D. purpureum var. pseudosessile*	MR273 (4637)	MH280022.1
*H. migratissimum*		KP167481.1	*D. purpureum var. pseudosessile*	MR273 (4446)	MH280023.1
*H. multicystogenum*	AS2	HQ141506.1	*D. quercibrachium*	NZ201B	HQ141479.1
*H. oculare*		HQ141497.1	*D. robusticaule*	5729-bai-2021	MW931857.1
*H. oculare*	DB4B	AM168079.1	*D. robusticaule*	5729-huang -2021	MW931856.1
*H. pallidum*	PPHU8	EU004605.1	** *D. robusticaule* **	**6700D**	**PQ237105**
*H. pallidum*	TNS-C-98	AM168103.1	** *D. macrosoriobrevipes* **	**6718S1-2B**	**PQ304778**
*H. parvimigratum*		KP167483.1	*D. robustum*	TNS-C-219	AM168064.1
*H. plurimicrocystogenum*		KP167475.1	*D. rosarium*	M45	AM168065.1
*H. pseudocandidum*	TNS-C-91	AM168107.1	*D. septentrionale*	IY49	AM168066.1
*H. pseudocolligatum*		KP167474.1	*D. septentrionale*	AK2	AM168067.1
*H. pseudoplasmodiofascium*		KP167482.1	*D. sphaerocephalum*	GR11	AM168068.1
*H. pseudoplasmodiomagnum*		KP167472.1	*D. valdivianum*		GQ496155.1
*H. racemiferum*		KP167476.1	*D. brevicaule*	SMA	
*H. radiatum*	M26B	MN338953.1	*Polysphondylium fuscans*	Sweden-11D	JX173877.1
*H. rotatum*	QC2C	HQ141501.1	*P. laterosorum*	AE4	AM168046.1
*H. stolonicoideum*	K12A	HQ141507.1	*P. patagonicum*		GQ496156.1
*H. tenuissimum*	TNS-C-97	AM168105.1	** *P. patagonicum* **	**6718-Z2**	**PQ236756**
*H. tenuissimum*		AY040339.1	** *P. patagonicum* **	**6718-zi**	**PQ236757**
*H. tikalense*	OH595	AM168106.1	*P. violaceum*	209	HQ141486.1
*H. tikalense*	HN1C1	HQ141509.1	*P. violaceum*	P6	AM168108.1
*H. unguliferum*		KP167473.1	*P. acuminatum*	OH500 SML	
*H. versatile*	Mad52	MN338954.1	*Coremiostelium polycephalum*	Landolt # 1130 SS3B	HQ141488.1
*H. violaceotypum*		KP167478.1	*C. polycephalum*	Landolt # 2132 B-9c	HQ141489.1
*H. recretum*	5756-1-17	MW857293.1	*C. polycephalum*	Landolt # 1675 GUAM	HQ141490.1
*H. multibrachiatum*	5916lun	MN752217.1	*C. polycephalum*	MY1-1	AM168056.1
*H. naviculare*	JC SMA		*Synstelium polycarpum*	VE1b	AM168057.1
*Tieghemostelium angelicum*	38B0	JF892716.1	*S. polycarpum*	OhioWILDS	AM168058.1
*T. dumosum*	OH602	JF892722.1	*Physarum polycephalum*		X13160.1

Newly generated sequences are shown in bold.

## Data Availability

The original contributions presented in the study are included in the article and Appendix A, further inquiries can be directed to the corresponding author.

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
