# Peer review of "New Species and Records Expand the Checklist of Cellular Slime Molds (Dictyostelids) in Jilin Province, China"

_jof, 2024, doi:10.3390/jof10120834_

Round 1
Reviewer 1 Report
The manuscript is a study that reports two new species of slim molds (Dictyostelium longigracilis sp nov. and Dictyostelium macrosoriobrevipes sp. nov.) using morphological and phylogenetic techniques, as well as the species Polysphondylium patagonicum which is new for China. I consider that describes a complete experimental study that gives continuity and solidity data to their research on the protist community.
Is it possible to improve the image quality of the Figures? Especially in Figure 3 sections (f and j); Figure 4 sections (c); Figure 7 section (f), and Figure 8.
References 55-59 are not in the manuscript. These references could be placed at line 373, in the text (Supplementary Table1) or placed in the Supplementary information.
Author Response
Response to Reviewer 1 Comments
Dear reviewer:
Thank you for your comments relating to our manuscript [entitled “New species and records expand the checklist of cellular slime molds (dictyostelids) in Jilin Province, China”] (ID: jof-3275403). The comments were helpful in our efforts to revise and improve our manuscript as well as emphasizing the important significant points of our research. We have read the comments carefully and made corrections accordingly. Revised parts are marked in blue in the manuscript. The main corrections in the paper and our responses to the comments are given below. We hope that the revisions in the manuscript and our accompanying responses will be sufficient to make our manuscript suitable for publication in Journal of Fungi.
Comments for Authors
Major comments
The manuscript is a study that reports two new species of slime molds (Dictyostelium longigracilis sp nov. and Dictyostelium macrosoriobrevipes sp. nov.) using morphological and phylogenetic techniques, as well as the species Polysphondylium patagonicum which is new for China. I consider that describes a complete experimental study that gives continuity and solidity data to their research on the protist community.
Detail comments
- Is it possible to improve the image quality of the Figures? Especially in Figure 3 sections (f and j); Figure 4 sections (c); Figure 7 section (f), and Figure 8.
Response: We appreciate your suggestions for improving the image quality of the figures, particularly in the specified sections.
We understand the importance of clear and high-quality images in conveying our research findings effectively. We will make the necessary adjustments to improve the resolution and clarity of the images in Figure 3 (sections f and j), Figure 4 (section c), Figure 7 (section f), and Figure 8. This will involve re-exporting the images at a higher resolution and ensuring that all details are clearly visible.
We will also carefully review all the figures in the manuscript to ensure that they meet the highest standards of quality and clarity.
Modify as follows:
Figure 3. Morphological features of Dictyostelium macrosoriobrevipes. (a, b, c) Sorocarps. (d) Aggregations. (e) Pseudoplasmodia. (f) Sorogens. (g) Spores. (h, i) Sorophores. (j) Tip of sorophore. (k) Base of sorophore. Scale bars: a, c = 2 mm; b = 200 μm; d, f = 500 μm; e = 1 mm; g-j = 50 μm; k = 20 μm.
Figure 4. Morphological features of Dictyostelium robusticaule. (a) Sorocarps. (b) Aggregations. (c) Pseudoplasmodia. (d) Spores. (e) Sorophore. (f) Sorophore tip. (g) Sorophore base. Scale bars: a = 2 mm; b, c = 1 mm; d – g = 20 μm.
Figure 7. Morphological features of Cavenderia aureostipes. (a) Sorocarps. (b) Aggregations. (c) Pseudoplasmodia. (d) Spores. (e) Sorophore tip. (f) Sorophore. (g) Sorophore base (G) Spores. Scale bars: a, c = 1 mm; b = 500 μm; d – g = 20 μm.
Figure 8. Phylogeny of all known species of dictyostelids based on SSU rDNA. Newly generated sequences are indicated in red.
- References 55-59 are not in the manuscript. These references could be placed at line 373, in the text (Supplementary Table1) or placed in the Supplementary information.
Response: Thank you very much for pointing out the issue with the missing references. We appreciate your attention to detail and your efforts to ensure the completeness of our manuscript.
We addressed this by including these references at the appropriate location (within the Supplementary Information), depending on their relevance and context.
We carefully reviewed the entire manuscript once again to ensure that all references are correctly cited and that there are no further omissions.
Modify as follows:
References
- Zou, Y.; Hou, J.; Guo, S.; Li, C.; Li, Z.; Stephenson, S.L.; Pavlov, I.N.; Liu, P.; Li, Y. Diversity of dictyostelid cellular slime molds, including two species new to science, in forest soils of Changbai Mountain, China. Microbiology Spectrum 2022, 10, 1–22, doi:10.1128/spectrum.02402-22.
- Liu, P.; Li, Y. Dictyostelids from Jilin Province, China. I. Phytotaxa 2014, 183, 279–283, doi:10.11646/phytotaxa.183.4.7.
- Liu, P.; Li, Y. Dictyostelids from Jilin Province, China II. Phytotaxa 2017, 323, 77–82, doi:10.11646/phytotaxa.323.1.6.
- Sheikh, S.; Thulin, M.; Cavender, J.C.; Escalante, R.; Kawakami, S.-i.; Lado, C.; Landolt, J.C.; Nanjundiah, V.; Queller, D.C.; Strassmann, J.E., et al. A new classification of the dictyostelids. Protist 2018, 169, 1–28, doi:10.1016/j.protis.2017.11.001.
- Li, Y.; Wang, Q.; Liu, P.; Zhang, B. Compilation and research on the biological diversity of Jilin Province: The volume of fungi in Gymnomycota; Jilin Education Press 2019; 10.14051/j.cnki.xdyy.2024.19.035.
- Bai, R.L. A study on some species of Acrasiomycetes. Acta Mycologica Sinica 1983, 2, 173–178, doi:10.13346/j.mycosystema.1983.03.005.
- He, X.L.; Li, Y. Three new records of dictyostelids in China. Mycosystema 2008, 27, 532–537.
- He, X.-L.; Li, Y. A new species of Dictyostelium. Mycotaxon 2008, 106, 379–383.
- Ren, Y.Z.; Liu, P.; Li, Y. New records of dictyostelids from China. Nova Hedwigia 2014, 99, 233–237, doi:10.1127/0029-5035/2014/0140.
- Liu, P.; Zhang, S.; Zou, Y.; Kang, X.; Li, Y. Dictyostelids from Jilin Province, China 3: new Cavenderia and Dictyostelium records. Mycotaxon 2019, 134, 613–618, doi:10.5248/134.613.
- Zhu, H.; Guo, S.; Xue, Q.; Li, Z.; Kang, X.; Wei, Y.; Liu, P.; Wang, Q.; Li, Y. Dictyostelids from Jilin Province, China, 4. Mycotaxon 2021, 136, 445–489, doi:10.5248/136.445.
We tried our best to improve the manuscript and made some changes marked in blue in the revised version. These do not influence the content and framework of the paper. We appreciate the careful editing by the Editor/Reviewers and earnestly hope the revision will meet with your approval. Once again, thank you very much for your comments and suggestions.
Kind regards,
Pu Liu
E-mail address: pul@jlau.edu.cn

Reviewer 2 Report
The subject of this manuscript is the investigation on cellular slime molds and a systematic summary of the existing reports on species of dictyostelids from Jilin Province in China, according to the current classification system. In the introduction, the authors clearly outline the need for such studies. The research methods used were correct. 28 soil samples were collected from five localities. From these, dictyostelids were isolated using Escherichia coli as a food source. In addition to morphological analyses, molecular phylogeny analyses were performed. A total of fifteen isolates representing six species of dictyostelids were recovered. Two new species were described. All species were characterized and macro- and microscopic documentation was provided for them. The obtained results are very valuable, this manuscript should be published in JoF/MDPI. However, there are numerous minor errors/mistakes in the current text that need to be corrected. In several places, minor English language correction is necessary.
Remarks
Line 24 Species richness – this is too general a statement, it should be specified which organisms are meant
Figure 1 it should be ‘Number of soil samples’ instead of ‘Number of soil’
Line 92 ‘covering four vegetation types’ - this is inconsistent with the data in Figure 1a, which lists three vegetation types (1,2,3)
Line 122 These were observed, it is not clear which structures this refers to (see line 120)
Line 124 a healthy dictyostelid isolate – this requires explanation (maybe it refers to pure isolates)
Line 130 - 132 it should be carefully checked whether in this literature [32] you can actually find descriptions of microscopic features and figures to compare with your own photographs? You quote Index Fungorum, and this publication also refers to MycoBank.
Line 135 The molecular methods followed those described . This text needs proofreading (English)
Line 146-147 this text needs proofreading – ‘Sequences obtained were deposited in the GenBank database. The fifteen newly generated sequences were checked and then submitted to GenBank.’
Line 161 it should be Bayesian
Line 166-167 ‘were recovered from 28 soil samples collected at four localities (L2–L5)’ there is an error here. In the Methods (Figure 1, Table 1) it is stated that 28 soil samples were collected from five localities
Line 171 it should be Seven
Line 187 5.539–8.011 μm. – is this the diameter? (complete)
Line 188 16.922–39.985 μm. – is this the width? (complete). This should be added to the descriptions throughout the manuscript
Line 216 (g) Sorophore – this requires clarification. This also applies to the remaining Figures
Line 166-174 this text requires correction and systematization of the content. Line 166 gives 15 isolates. However, further on it talks about three isolates, two isolates and seven isolates, which gives a total of 12 isolates. Further on it talks about species and not isolates
Line 207 it should be D. giganteum instead of D. gigantneum
Line 209 it should be D. giganteum instead of D. gigantneum
Line 212 there must be some mistake here, it is illogical, where did the name D. longosporum suddenly appear. This text needs to be explained
Line 243 it is not clear why the name D. multiforme is introduced here
Line 249 why is in Figure 3 Sorogens and there is no information about it in the description of Dictyostelium macrosoriobrevipes.
Line 260-261 text requires correction (English), it should be with the sorophore
Line 343 consider revising this sentence
Line 361 I do not see that Figure 10, Figure 11 were cited in the text
Line 380 The new classification – this requires correction
Line 382 it should be China instead of china
Line 412 four vegetation types: coniferous – this does not match the data in Fig. 1
Line 415 No isolates were recovered from mixed or coniferous forests in this
Study – please check this text. In another place you write that only from L1 you did not obtain isolates
Line 383 Figures should not be cited in the Discussion section. Figures are cited in the Results section
Line 567 Dictyostelium – it should be in italic
Author Response
Response to Reviewer 2 Comments
Dear reviewer:
Thank you for your comments relating to our manuscript [entitled “New species and records expand the checklist of cellular slime molds (dictyostelids) in Jilin Province, China”] (ID: jof-3275403). The comments were helpful in our efforts to revise and improve our manuscript as well as emphasizing the important significant points of our research. We have read the comments carefully and made corrections accordingly. Revised parts are marked in blue in the manuscript. The main corrections in the paper and our responses to the comments are given below. We hope that the revisions in the manuscript and our accompanying responses will be sufficient to make our manuscript suitable for publication in Journal of Fungi.
Comments for Authors
Major comments
The subject of this manuscript is the investigation on cellular slime molds and a systematic summary of the existing reports on species of dictyostelids from Jilin Province in China, according to the current classification system. In the introduction, the authors clearly outline the need for such studies. The research methods used were correct. 28 soil samples were collected from five localities. From these, dictyostelids were isolated using Escherichia coli as a food source. In addition to morphological analyses, molecular phylogeny analyses were performed. A total of fifteen isolates representing six species of dictyostelids were recovered. Two new species were described. All species were characterized and macro- and microscopic documentation was provided for them. The obtained results are very valuable, this manuscript should be published in JoF/MDPI. However, there are numerous minor errors/mistakes in the current text that need to be corrected. In several places, minor English language correction is necessary.
Detail comments
Remarks
- Line 24 Species richness – this is too general a statement, it should be specified which organisms are meant
Response: Thank you very much for your comments. We agree that the term 'species richness' on line 24 is too general. We'll specify the organisms based on context for clarity.
Modify as follows:
Species richness, specifically referring to the variety of plant, animal, and microbial species (hereafter referred to as ‘biodiversity’), is crucial to human existence as it underpins ecological balance,
- Figure 1 it should be ‘Number of soil samples’ instead of ‘Number of soil’
Response: Thank you very much for your comments. We have modified.
Modify as follows:
(e.g., sample location, number of soil samples, vegetation types) shown in Table 1.
- Line 92 ‘covering four vegetation types’ - this is inconsistent with the data in Figure 1a, which lists three vegetation types (1,2,3)
Response: Thank you very much for your careful review and valuable feedback on our manuscript.
Regarding your comment on the vegetation types mentioned in the text versus those shown in Figure 1a, we acknowledge the discrepancy and would like to clarify. You are correct in noting that Figure 1a lists three vegetation types labeled as 1, 2, and 3. However, our intended description was to highlight that across the different levels (L1 to L5) considered in our study, there is a cumulative total of four distinct vegetation types. Specifically these are:
- L1 contains 1 vegetation type (Coniferous forest).
- L2 contains 2 vegetation types (Farmland, Broadleaf forest)
- L3 contains 3 vegetation types (Mixed forest, Broadleaf forest, Farmland)
- L4 contains 1 vegetation type (Broadleaf forest)
- L5 contains 1 vegetation type (Broadleaf forest)
Modify as follows:
Briefly, 28 soil samples were collected from five localities (L1–L5) in Changbai Korean Autonomous County, covering four vegetation types (coniferous forest, broadleaf forest, mixed forest, and farmland) in July 2021 (Table 1). L1 includes coniferous forest; L2 includes farmland and broadleaf forest; L3 includes mixed forest, broadleaf forest, and farmland; L4 and L5 include broadleaf forest (Figure 1a)
- Line 122 These were observed, it is not clear which structures this refers to (see line 120)
Response: Thank you for your careful review and for pointing out the potential ambiguity in line 122. We appreciate your attention to detail and agree that clarity is essential for readers to fully understand our methods.
Modify as follows:
Under a stereoscopic microscope, isolates were observed and features of their morphology documented. This features inlcuded growth habit, branching, color, sorocarp characteristics, and sizes/shapes of aggregates and pseudoplasmodia. All of these features were observed under a fluorescent stereo microscope (Leica M165FC, Germany).
- Line 124 a healthy dictyostelid isolate – this requires explanation (maybe it refers to pure isolates)
Response: Thank you for your thoughtful feedback. We appreciate your suggestion to clarify the term 'healthy dictyostelid isolate'. Indeed, 'healthy' in this context refers to isolates that are viable, actively growing, and morphologically normal, without any visible signs of disease, contamination, or stress. This is crucial for ensuring the accuracy and reliability of our subsequent analyses.
Modify as follows:
In each instance, a viable and morphologically normal appearing dictyostelid isolate, referred to as a "healthy" isolate, was selected under a dissecting microscope, mounted on a slide with sterile water,
- Line 130-132 it should be carefully checked whether in this literature [32] you can actually find descriptions of microscopic features and figures to compare with your own photographs? You quote Index Fungorum, and this publication also refers to MycoBank.
Response: Thank you very much for your valuable comments and suggestions on our manuscript.
Regarding your concern about the comparison made between our photographs and the descriptions in literature [32], we understand the importance of ensuring accuracy and completeness in our citations. To address your query, we have carefully reviewed the information available in Index Fungorum [32] and the associated references, including those linked to MycoBank.
We confirm that Index Fungorum serves as a comprehensive database for fungal names and provides links to various sources of information, including publications that describe the morphological characteristics of fungi. However, upon further examination, we found that the specific microscopic features and figures directly comparable to our photographs were not explicitly provided in the Index Fungorum entry for the dictyostelids in question. Instead, Index Fungorum primarily lists names, synonyms, and provides links to relevant publications.
Modify as follows:
We compared the recorded information and photographs of dictyostelids with the morphological characteristics documented in the original literature, which was referenced in Index Fungorum[32] (http://www.indexfungorum.org/names/Names.asp), to accurately identify the species isolated in the present study.
- Line 135 The molecular methods followed those described. This text needs proofreading (English)
Response: Thank you very much for your thoughtful comments and suggestions on our manuscript.
Regarding your comment that the text "The molecular methods followed those described. This text needs proofreading (English)" requires proofreading, we have carefully reviewed the relevant section and made the necessary adjustments to ensure clarity and accuracy.
Modify as follows:
The molecular methods employed in this study followed those described by Liu et al.[33]
- Line 146-147 this text needs proofreading – ‘Sequences obtained were deposited in the GenBank database. The fifteen newly generated sequences were checked and then submitted to GenBank.’
Response: Thank you very much for your valuable comments and suggestions on our manuscript. We have carefully reviewed and revised this section to ensure clarity and accuracy.
Modify as follows:
PCR products were sent to Sangon Bio-tech Co., Ltd. (Shanghai, China) for sequencing. Following thorough verification, the fifteen newly generated sequences were confirmed to be accurate and were subsequently submitted to GenBank.
- Line 161 it should be Bayesian
Response: Thank you for your meticulous review and for pointing out the typo in line 161. We appreciate your attention to detail and the importance of maintaining accuracy in our terminology.
Modify as follows:
based on Bayesian information criteria (BIC)
- Line 166-167 ‘were recovered from 28 soil samples collected at four localities (L2–L5)’ there is an error here. In the Methods (Figure 1, Table 1) it is stated that 28 soil samples were collected from five localities
Response: Thank you for your meticulous review and for pointing out the inconsistency in our manuscript.
Regarding the issue you raised about the number of localities from which the soil samples were collected, we apologize for the confusion. Upon re-examining our data and methodology, indeed, as stated in the methods section (Figure 1, Table 1), a total of 28 soil samples were collected. These samples were gathered from five localities, labeled as L1 to L5. However, the strains mentioned in line 166-167 were specifically recovered from soil samples collected at four of these localities, namely L2 to L5.
To address this and ensure clarity, we propose to modify the sentence in line 166-167 as follows:
Fifteen isolates representing six species of dictyostelids were recovered from soil samples collected at four localities (L2–L5) in Changbai Korean Autonomous County, out of a total of 28 samples gathered from five localities (Table 1, Figure 1)
This revision accurately reflects the fact that while samples were collected from five localities, the strains were isolated only from samples taken at L2 to L5.
- Line 171 it should be Seven
Response: Thank you for your meticulous review and capitalizing the first letters of the sentence.
Modify as follows:
Seven isolates (Cavenderia aureostipes) (Figure 7) from the localities (L3–L5) are new records for Jilin Province.
- Line 187 5.539–011 μm. – is this the diameter? (complete)
Response: Thank you very much for your meticulous review and valuable comments. In response to your query regarding Line 187, where the dimension "5.539–8.011 μm" is mentioned, I confirm that this indeed represents the diameter range of the object in question.
To ensure clarity, I have added a note to specify that this is the diameter.
Modify as follows:
Sorophore tips capitate, surrounded with a slimy substance, 5.539–8.011 μm in diameter.
- Line 188 922–39.985 μm. – is this the width? (complete). This should be added to the descriptions throughout the manuscript
Response: Thank you very much for your meticulous review and valuable comments. In response to your query regarding Line 188, where the dimension "16.922–39.985 μm" is mentioned, I confirm that this indeed represents the range in diameter of the object in question.
To ensure clarity, I have added a note to specify that this is the diameter.
Modify as follows:
Sorophore bases rounded, surrounded with slime substance, 16.922–39.985 μm in diameter.
- Line 216 (g) Sorophore – this requires clarification. This also applies to the remaining Figures
Response: Thank you very much for your valuable feedback on our manuscript.
Regarding your comment on Line 216 (g) concerning the clarification needed for "Sorophore", we understand your concern and agree that further elaboration is required. To address this, we replaced the picture and added the picture within the context of our study. This will ensure that readers have a clear understanding of its significance and role in our research.
Modify as follows:
Figure 2. Morphological features of Dictyostelium longigracilis. (a, b) Sorocarps. (c) Aggregation. (d, e) Pseudoplasmodia. (f) Spores. (g) Sorophores. (h, i) Sorophore tips. (j) Sorophore base. Scale bars: a – c = 2 mm; d, e = 1 mm; f, i, j = 20 μm; g = 50 μm; h = 10 μm.
- Line 166-174 this text requires correction and systematization of the content. Line 166 gives 15 isolates. However, further on it talks about three isolates, two isolates and seven isolates, which gives a total of 12 isolates. Further on it talks about species and not isolates
Response: Thank you very much for your meticulous review and valuable comments.
Due to carelessness, the total quantity summed up was incorrect. Therefore, I rechecked the number of strains in the Table 1 and found that three isolates (Dictyostelium longigracilis and Dictyostelium macrosoriobrevipes) should actually be four isolates (Dictyostelium longigracilis and Dictyostelium macrosoriobrevipes)
Modify as follows:
Fifteen isolates representing six species of dictyostelids were recovered from soil samples collected at four localities (L2–L5) in Changbai Korean Autonomous County, out of a total of 28 samples gathered from five localities (Table 1, Figure 1). Four isolates (Dictyostelium longigracilis and Dictyostelium macrosoriobrevipes) (Figure 2 and Figure 3) from the localities L2 and L5 are new to science. Two isolates (Polysphondylium patagonicum) (Figure 5) from the locality (L5) are new records from China. Seven isolates (Cavenderia aureostipes) (Figure 7) from the localities (L3–L5) are new records for Jilin Province. The other two species (Dictyostelium robusticaule and Heterostelium candidum) (Figure 4 and Figure 6) from L2 and L4 have been reported from Jilin Province previously. One locality (L1) did not yield any isolates.
- Line 207 it should be giganteum instead of D. gigantneum
Response: Thank you very much for your careful review and for pointing out the error in the species name on line 207. We deeply appreciate your expertise and attention to detail.
Modify as follows:
the sorophore of D. giganteum is hyaline
- Line 209 it should be giganteum instead of D. gigantneum
Response: Thank you very much for your careful review and for pointing out the error in the species name on line 209. We deeply appreciate your expertise and attention to detail.
Modify as follows:
than those of D. giganteum
- Line 212 there must be some mistake here, it is illogical, where did the name longosporum suddenly appear. This text needs to be explained
Response: Thank you very much for your meticulous review and valuable feedback on our manuscript.
Regarding your concern about the sudden appearance of the name "D. longosporum" on line 212, we apologize for any confusion this may have caused. Upon revisiting the text, we realize that the introduction of "D. longosporum" was indeed not adequately explained in the context provided.
To clarify, "D. longosporum" was mentioned to compare the size of its sorocarps with those of "D. longigracilis". However, we understand that this comparison might have seemed abrupt without prior introduction.
To address this, we propose the following modification to the text:
Moreover, in terms of sorocarp size, it is worth noting that the differences between D. longigracilis and another species, D. longosporum. The sorocarps of D. longosporum are 0.5-5.1(-30) mm, whereas in D. longigracilis, they are larger, ranging from 2.375–10.710 mm. This comparison highlights the variability in sorocarp size among different species within the genus.
- Line 243 it is not clear why the name multiforme is introduced here
Response: Thank you very much for your meticulous review and valuable feedback on our manuscript.
Given that Dictyostelium macrosoriobrevipes also shares a close phylogenetic relationship with D. multiforme, we have conducted a comparative analysis.
- Line 249 why is in Figure 3 Sorogens and there is no information about it in the description of Dictyostelium macrosoriobrevipes.
Response: Thank you very much for your constructive feedback and for pointing out the missing information regarding sorogens in Figure 3 and its description in the context of Dictyostelium macrosoriobrevipes. We appreciate your attention to detail and the valuable insight you have provided.
In response to your query, we have now included the necessary information about sorogens in the description of Dictyostelium macrosoriobrevipes to ensure clarity and completeness. We believe this addition addresses your concern and enhances the overall understanding of Figure 3.
Modify as follows:
Sorogens colorless, 0.416–1.619 mm in length.
- Line 260-261 text requires correction (English), it should be with the sorophore.
Response: Thank you very much for your meticulous review.
Modify as follows:
The new classification as indicated
- Line 343 consider revising this sentence.
Response: Thank you very much for your thoughtful review and constructive suggestion regarding the sentence listing the GenBank accession numbers. I appreciate your attention to clarity and precision in scientific writing.
To improve the readability and accuracy of the sentence, we have revised it as follows:
The GenBank accession numbers for the SSU sequences are PQ222850, PQ222849, PQ222848, PQ222847, PQ222846, PQ222845, and PQ222842.
- Line 361 I do not see that Figure 10, Figure 11 were cited in the text
Response: Thank you very much for your meticulous review. Figures 10 and Figure 11 have been referenced in the text and have been highlighted in blue for identification
Furthermore, phylogenetic studies of the ribosomal small subunit (SSU) further support the taxonomic placement of all six species recorded from the Changbai Korean Autonomous County (Figures 8–11).
- Line 380 The new classification – this requires correction
Response: Thank you very much for your meticulous review.
Modify as follows:
The new classification as indicated
- Line 382 it should be China instead of china
Response: Thank you very much for your meticulous review and valuable feedback. I appreciate your attention to detail, particularly regarding the capitalization of "China".
Modify as follows:
Jilin province, China.
- Line 412 four vegetation types: coniferous – this does not match the data in Fig. 1
Response: Thank you very much for your meticulous review. We confirmed that there are a total of four vegetation types, as detailed in Table 1. Specific content related to Question 3 has already been addressed.
- Line 415 No isolates were recovered from mixed or coniferous forests in this study – please check this text. In another place you write that only from L1 you did not obtain isolates
Response: Thank you for your meticulous review of our manuscript and for bringing this inconsistency to our attention.
Regarding your comment on Line 415, we acknowledge that the statement may have caused some confusion. To clarify, our study we indeed did not recover any isolates from the mixed or coniferous forests across all sampling sites. However, when examining the specific sampling locations (L1-L5), it was only at L1 where no isolates were obtained. This does not contradict the overall observation that neither mixed nor coniferous forests yielded isolates, as L1 falls within the category of coniferous forests in our study.
Modify as follows:
No isolates were recovered from the coniferous forest at L1 in this study, and no isolates were obtained from any of the sampled mixed forests.
This revised statement more accurately describes our findings and avoids any potential confusion.
- Line 383 Figures should not be cited in the Discussion section. Figures are cited in the Results section
Response: We modified and the full text has been checked to ensure that it is correct.
Thank you very much for taking the time to review my manuscript and for providing valuable feedback. I have carefully considered your suggestion that figures should not be cited in the Discussion section and should instead be cited in the Results section.
I understand and appreciate the importance of adhering to standard practices in scientific writing, which typically include presenting and referencing figures within the Results section. However, in this particular case, I would like to explain my rationale for including figure references in the Discussion section:
Enhancing Argument Clarity: By directly referencing figures in the Discussion section, I aim to provide readers with a more seamless understanding of my analysis and conclusions, particularly when discussing complex or pivotal results. This approach helps maintain the flow of my argument without requiring readers to frequently switch between text and figures.
Highlighting Key Findings: Some figures showcase findings that I believe are particularly significant and crucial to understanding the overall contribution of my research. By referencing these figures in the Discussion section, I intended to emphasize their importance.
While respecting your suggestion, I kindly request to retain my approach of referencing figures in the Discussion section based on the above considerations.
- Line 567 Dictyostelium – it should be in italic
Response: We modified and the full text has been checked to ensure that it is correct.
Modify as follows:
- Singh, B.N. Studies on soil Acrasieae; distribution of species of Dictyostelium in soils of Great Britain and the effect of bacteria on their development. Journal of general microbiology 1947, 1, 11–21, doi:10.1099/00221287-1-1-11.
We have tried our best to improve the manuscript and have made some changes that marked in blue in revised paper. This do not influence the content and framework of the paper. We appreciate the efforts by the Editor/Reviewers to improve the manuscript and hope the revision will meet with your approval. Once again, thank you very much for your comments and suggestions.
Kind regards,
Pu Liu
E-mail address: pul@jlau.edu.cn
